# The phosphatidylserine targeting antibody bavituximab plus pembrolizumab in unresectable hepatocellular carcinoma: a phase 2 trial

Immune checkpoint inhibitors targeting PD-1/L1 have modest efficacy in hepatocellular carcinoma as single agents. Targeting membranous phosphatidylserine may induce pro-inflammatory and -immune stimulating effects that enhance immunotherapy activity. This hypothesis was tested in a single-arm phase 2 trial evaluating frontline bavituximab, a phosphatidylserine targeting antibody, plus pembrolizumab (anti-PD-1) in patients with unresectable hepatocellular carcinoma (NCT03519997). The primary endpoint was investigator-assessed objective response rate among evaluable patients, and secondary end points included progression-free survival, incidence of adverse events, overall survival, and duration of response. Among 28 evaluable patients, the confirmed response rate was 32.1%, which met the pre-specified endpoint, and the median progression-free survival was 6.3 months (95% CI, 1.3–11.3 months). Treatment related-adverse events of any grade occurred in 45.7% of patients, with grade 3 or greater adverse events in 14.3% of patients. Adverse events of any cause were observed in 33 patients (94.3%), with grade 3 or greater adverse events in 11 patients (31.4%). Prespecified exploratory analyses of baseline tumor specimens showed that a depletion of B cells, and the presence of fibrotic tissue and expression of immune checkpoints in stroma was associated with tumor response. These results suggest that targeting phosphatidylserine may lead to synergistic effects with PD-1 blockade without increasing toxicity rates, and future studies on this therapeutic strategy may be guided by biomarkers characterizing the pre-treatment tumor microenvironment.

Hepatocellular carcinoma (HCC) is the most common liver cancer and the fourth most frequent cause of cancer-related deaths worldwide[1]. Given a high recurrence rate among patients who undergo resection and a high frequency of advanced stage disease at the time of diagnosis, effective systemic treatments for HCC are needed to combat its growing burden of disease. While tyrosine kinase inhibitors targeting VEGF signaling were previously the mainstay of HCC therapies, immune checkpoint inhibitor combinations have emerged as the preferred frontline treatments for advanced HCC based on superior anti-tumor effects and meaningful survival benefits. In the phase 3 IMbrave150 study, atezolizumab (anti-PD-L1) plus bevacizumab in the first line setting demonstrated an unprecedented objective response

e-mail: David.hsieh@utsouthwestern.edu

rate (ORR) of 30% and overall survival (OS) benefit versus sorafenib[2]. Similarly, the phase 3 HIMALAYA trial showed that durvalumab (anti-PD-L1) plus tremelimumab (anti-CTLA4) was superior to sorafenib in both OS benefit and ORR (20.1% versus 5.1%)[3]. These studies demonstrate the clinical benefit of rationally designed combination immunotherapy regimens in a subset of patients with HCC. However, an increased risk of specific toxicities associated with each of these regimens limits their utility. Specifically, atezolizumab plus bevacizumab is associated with a greater risk of variceal bleeding, cardiovascular events, and clinically significant proteinuria which is attributed to the on-target effects of bevacizumab[2]. Tremelimumab plus durvalumab is associated with a greater risk of serious treatment-related toxicities compared to durvalumab alone (17.5% versus 8.2%)[3]. Thus, a treatment regimen with similar efficacy but an improved safety profile compared to approved immune checkpoint inhibitor combinations may have greater utility in patients with HCC who frequently have concurrent morbidities that may limit tolerance to adverse events.

Phosphatidylserine is a cytoplasmic-facing anionic phospholipid which has multiple essential functions in cellular membranes[4,5]. Externalization of phosphatidylserine occurs as a result of apoptosis or cellular stress, but is also frequently observed in cancer cells and within stromal cells[4]. Exposed phosphatidylserine and phosphatidylserine dependent-receptors include the Tyro3, Axl, and Mer (TAM) family of negative immune regulators may facilitate cancer immune escape[4]. Preclinical evidence indicates that antibody-based targeting of phosphatidylserine promotes pro-inflammatory pathways, recruitment of tumoricidal macrophages, dendritic cell maturation, and potent T-cell immunity in multiple cancer models[6]. Bavituximab is a genetically engineered immunoglobulin gamma 1 (IgG1) chimeric (human/mouse) antibody that targets phosphatidylserine by binding to the phosphatidylserine-binding protein β2-glycoprotein 1 (β2GP1)[6]. While bavituximab has been demonstrated to be well tolerated as a single agent and in combination with chemotherapy, its use as a single agent in a phase I study of refractory advanced solid cancers and in combination with docetaxel in a phase 3 trial for advanced non-small-cell lung cancer did not show improved clinical benefit[7–9]. In the frontline setting for HCC, bavituximab plus sorafenib was also not associated with a meaningful increase in ORR (5.3%)[10]. Importantly, prior use of bavituximab was not guided by its recently discovered immunogenic mechanism of action. The pleiotropic pro-inflammatory and -immune stimulating effects of bavituximab on tumor cells, immune cells, and stroma suggests that it may enhance the activity of immune checkpoint inhibitors. This is supported by preclinical evidence that targeting phosphatidylserine enhances either PD-1 or CTLA-4 blockade in multiple cancer models[11–13].

Pembrolizumab, an anti-PD-1 antibody, has modest anti-tumor activity in HCC. In the non-randomized phase 2 KEYNOTE-224 study, the ORR was 16% in patients previously treated with sorafenib (Cohort 1) and 16% in patients with no prior systemic therapy (Cohort 2)[14,15]. In the randomized, double-blind, phase 3 KEYNOTE-240 study testing pembrolizumab versus placebo in the second line setting, OS and PFS did not reach statistical significance per specified criteria, but there was an improvement in ORR (18.3% versus 4.4%)[16]. In the randomized, double-blind, phase 3 KEYNOTE-394 study testing pembrolizumab plus best supportive care versus best supportive care in Asia, pembrolizumab was associated with an improved OS, PFS, and ORR (13.7% versus 1.3%)[17]. Whether the efficacy of PD-1 blockade can be augmented by targeting novel regulators of the innate immune system in HCC beyond inhibitors of VEGF or CTLA-4 remains unclear.

Here, we investigate whether bavituximab may enhance the efficacy of pembrolizumab in HCC in a multicenter single-arm two-stage phase 2 trial of bavituximab plus pembrolizumab for unresectable HCC in patients who had not received prior systemic therapies. The primary endpoint was investigator determined confirmed ORR among evaluable patients. Treatment with bavituximab plus pembrolizumab

in this study met its prespecified endpoint, demonstrating a confirmed ORR of 32.1% (9 out of 28 evaluable patients). Treatment-related adverse events of any grade occurred in 45.7% of patients, with grade 3 or greater adverse events in 14.3% of patients. Depletion of B cells, intratumoral fibrosis, stromal immune checkpoint expression, and immune-rich inflammatory tumors were associated with tumor response.

## Results

### Patient enrollment and characteristics

From 25 June 2018 to 2 March 2022, 42 patients were screened for eligibility at a National Cancer Institute-designated cancer center (UT Southwestern Harold C. Simmons Comprehensive Cancer Center) and a county safety-net hospital system (Parkland Health), out of which 35 patients were enrolled (Supplementary Fig. 1). Enrolled patients who were not evaluable because they did not complete radiographic evaluation after starting treatment included 2 patients who withdrew consent, 1 patient who withdrew from the study after receiving radiation to a target lesion during their first cycle, 1 patient withdrawn following complications of an elective procedure, and 3 patients removed per investigator for worsening concomitant illness (1 patient who developed suspected immune-related colitis, 1 patient who developed hepatic decompensation during their first cycle, and 1 patient who developed non-islet cell hypoglycemia).

Based on a minimax two-stage design, 15 evaluable patients were enrolled in the first stage where the threshold to meet 3 or more confirmed objective responses was met allowing for the enrollment of an additional 13 evaluable patients in the second stage. Patients included in the primary analysis had a median age of 64 (IQR: 60–67), were primarily male (85.7%), frequently associated with HCV cirrhosis (71.4%), and frequently received prior locoregional therapy (60.7%) (Supplementary Table 1). These demographics are consistent with characteristics of patients with HCC in the broader US population[18,19]. Due to the enrollment of patients across clinical sites serving diverse demographics, 50% of the evaluable patients were Black, and 7.1% were Hispanic.

### Efficacy

Using RECIST 1.1 criteria, the investigator determined ORR among the 28 evaluable patients was 32.1%, including 2 complete response (CR) and 7 partial responses (PR), which met the prespecified primary endpoint of 8 or greater confirmed objective responses (Fig. 1a). The disease control rate including CRs, PRs, and stable disease (SD) was 64.3%. The median time to response was 2.1 months (range, 1.9–12.7 months), and the median duration of response was 13.3 months, with 4 responses ongoing at the time of data cutoff. The 6-month progression-free survival (PFS) rate was 57.1% (95% CI 38.8–75.4) and the median PFS was 6.3 months (95% CI, 1.3–11.3 months) (Fig. 1b). The median follow-up time was 28.5 months using the reverse Kaplan-Meier method. Noting that statistical power is limited given small sample sizes, subgroup analyses showed that objective tumor responses were not associated with clinical characteristics including age, race, sex, AFP levels, history of prior locoregional therapy, extrahepatic disease, or macrovascular invasion (Supplementary Fig. 2).

### Safety

Patients who received at least one dose of drug were included in the safety analysis as prespecified in the trial protocol. Among 35 patients included in the safety analysis, adverse events of any cause were observed in 33 patients (94.3%) and serious adverse events of any cause were observed in 7 patients (20%). Adverse events of any cause that were grade 3 or greater occurred in 11 patients (31.4%). Treatment-related adverse events occurred in 16 patients (45.7%) and serious adverse events attributed to study treatment occurred in 4 patients

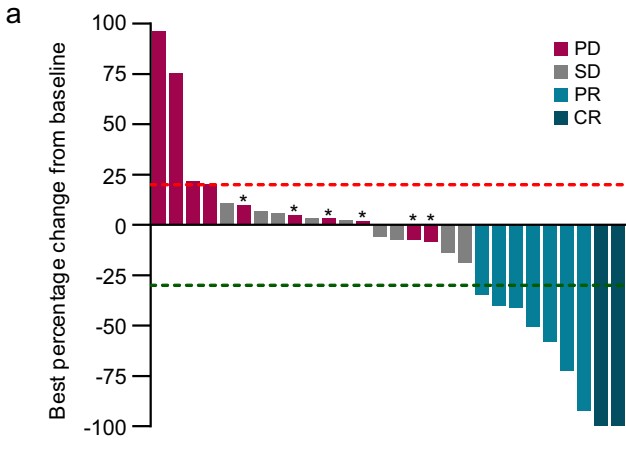

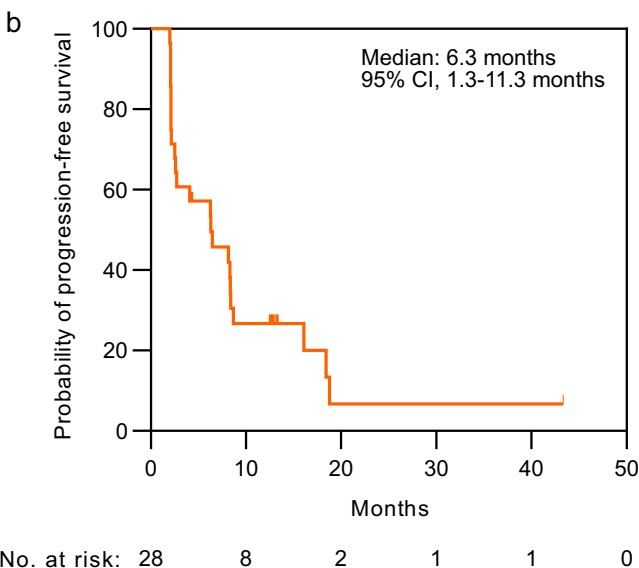

**Fig. 1 | Response rates and progression-free survival with bavituximab plus pembrolizumab in HCC. a** Waterfall plot depicts best percentage change in tumor dimensions from baseline per RECIST 1.1 among evaluable patients (*n* = 28). Asterisks indicates patients with new lesions demonstrating PD. ORR objective response rate, PR partial response, CR complete response, SD stable disease, PD progressive disease. **b** Kaplan-Meier curve shows the probability of progression-free survival over time among evaluable patients. Source data are provided as a Source data file.

(11.4%). Treatment-related adverse events that were grade 3 or greater occurred in 5 patients (14.3%). The most common treatment-related adverse event of any grade was diarrhea (17.1%), followed by rash (17.1%), alanine aminotransferase increase (11.4%), aspartate aminotransferase increase (11.4%), and chills (8.6%) (Table 1). Grade 3 and 4 treatment-related adverse events included diarrhea (5.7%), alanine aminotransferase increase (2.9%), aspartate aminotransferase increase (2.9%), and colitis (2.9%) (Table 1). Treatment interruptions due to adverse events occurred in 6 patients, and 1 patient discontinued therapy permanently for grade 4 diarrhea per the trial protocol for suspected high-grade immune-related toxicity.

## Genomic correlates of response
Genomic data from clinical sequencing of tissue specimens or circulating tumor DNA was available for 19 patients evaluable for the primary ORR endpoint. As a prespecified exploratory endpoint, we assessed whether molecular alterations were associated with response to bavituximab and pembrolizumab. Alterations involving *TP53* (63%),

## Table 1 | Treatment-related adverse events

| Event | Any Grade, *n* (%) | Grade 3 or 4, *n* (%) |
| --- | --- | --- |
| Diarrhea | 6 (17.1) | 2 (5.7) |
| Rash | 6 (17.1) | 1 (2.9) |
| Alanine aminotransferase increase | 4 (11.4) | 0 (0) |
| Aspartate aminotransferase increase | 4 (11.4) | 1 (2.9) |
| Chills | 3 (8.6) | 0 (0) |
| Arthritis | 2 (5.7) | 0 (0) |
| Fatigue | 2 (5.7) | 0 (0) |
| Platelet count decrease | 2 (5.7) | 0 (0) |
| Pruritus | 2 (5.7) | 0 (0) |
| Dyspnea | 2 (5.7) | 0 (0) |
| Pneumonitis | 1 (2.9) | 1 (2.9) |
| Abdominal pain | 1 (2.9) | 0 (0) |
| Albumin decrease | 1 (2.9) | 0 (0) |
| Bilirubinemia | 1 (2.9) | 0 (0) |
| Creatinine increase | 1 (2.9) | 0 (0) |
| Fever | 1 (2.9) | 0 (0) |
| Low Appetite | 1 (2.9) | 0 (0) |
| Mucositis oral | 1 (2.9) | 0 (0) |
| Pruritis | 1 (2.9) | 0 (0) |
| TSH increased | 1 (2.9) | 0 (0) |

*TERT* promoter (58%), *CTNNB1* (37%), and *ARID1A* (21%) were the most frequent somatic mutations observed (Fig. 2a). Other genes with recurrent alterations in the cohort included *EGFR* (11%), and *CCNE1* (11%). There were no statistically significant associations with objective responses with specific gene alterations including mutant *TP53* (41.7% vs 14.3%, two-sided Fisher's exact test, *p* = 0.33), wildtype *CTNNB1* (41.7% vs 14.3%, two-sided Fisher's exact test, *p* = 0.33), and *TERT* promoter (36.4% vs 25%, two-sided Fisher's exact test, *p* = 0.66) (Fig. 2b). Due to the limited sample size, this study is underpowered to detect associations between tumor response and less prevalent alterations.

## Spatial proteomic profiling
Pre-treatment archival tissue specimens were available from 20 patients for multiplex proteomic profiling of a panel of 52 immuno-oncology-related proteins using the GeoMx Digital Spatial Profiling (NanoString) technology. As a prespecified exploratory endpoint, we assessed whether proteomic features of baseline tissue specimens were associated with response to bavituximab and pembrolizumab. Segmentation based on cell type markers within a region of interest (ROI) in each tissue specimen was used to demarcate areas enriched with immune cells (CD45+), tumor cells (Pan-CK+ or morphologically tumor cells in H&E staining of serial section lacking positivity of the cell type markers), and stroma (SMA+) (Supplementary Fig. 3). Differential expression of immune-oncology proteins was assessed between patients with objective responses (CR or PR) versus patients without tumor response (SD or PD). Among the immune cell-enriched areas, CD20, a B cell-specific marker, was lower in the responders compared to the non-responders (Fig. 3). Within the stroma-enriched areas, SMA expression was higher in non-responders, while PD-L1, PD-L2, IDO1, and Bcl-2 expression was higher in responders (Fig. 3). There was no statistically significant differential protein expression in association with the objective response in the tumor cell-enriched areas (Fig. 3).

## RNA-based tumor microenvironment (TME) biomarker
RNA gene signatures characterizing immune phenotypes in the TME may be predictive of ICI efficacy[20–22]. As a prespecified exploratory

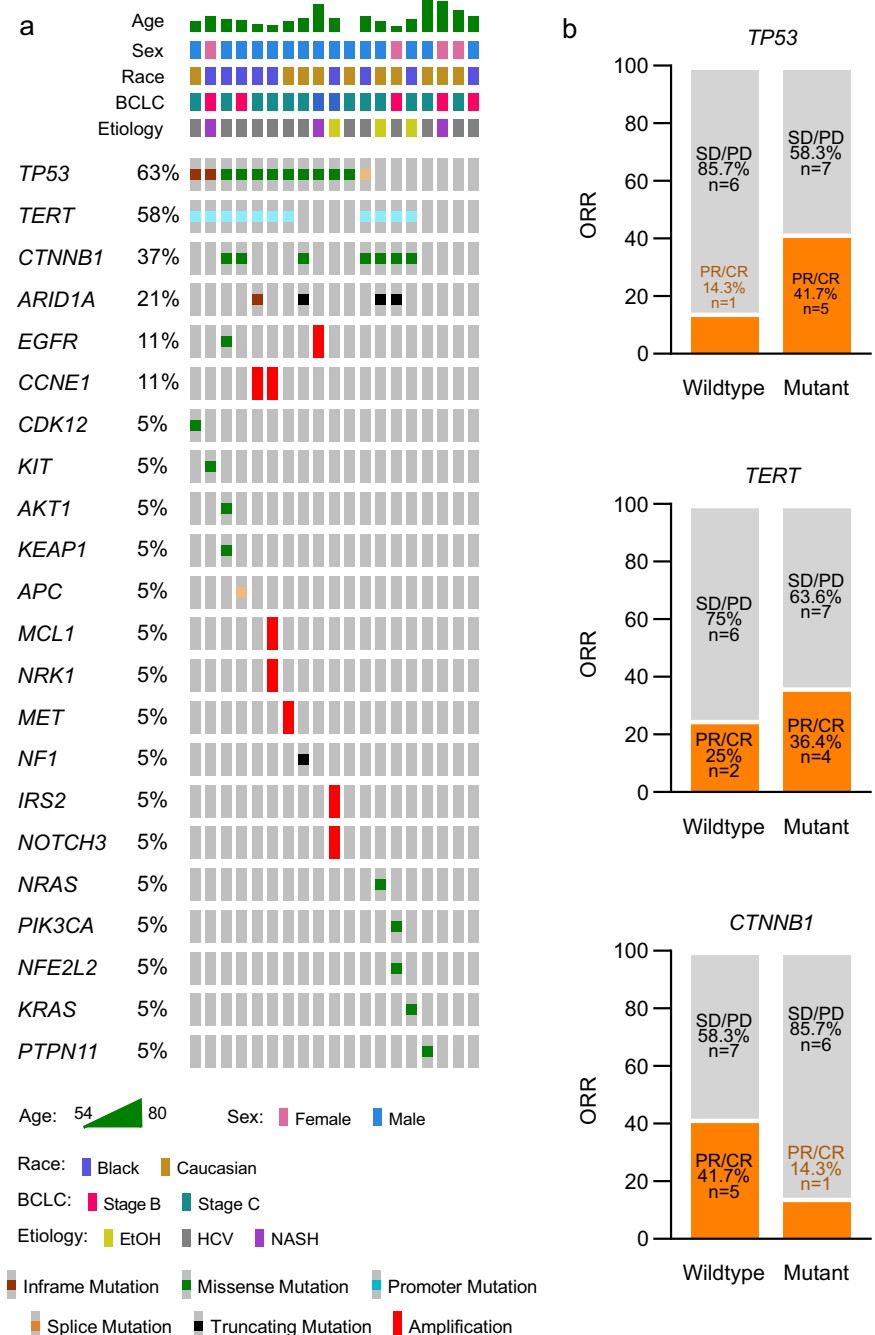

**Fig. 2 | Genomic profiling of evaluable patients treated with bavituximab plus pembrolizumab. a** Oncoprint plot depicts the landscape and distribution of genomic alterations among evaluable patients (*n* = 19) who underwent clinical sequencing. Barcelona clinic liver cancer, BCLC. **b** Stacked bar graphs depict proportion of patients with objective tumor responses (PR or CR) among patients with *TERT* promoter, *TP53*, or *CTNNB1* alterations. ORR objective response rate, PR partial response, CR complete response, SD stable disease, PD progressive disease. Fisher's exact tests were used to assess associations between tumor response and molecular. Source data are provided as a Source data file.

endpoint, we assessed whether RNA-based features of baseline tissue specimens were associated with response to bavituximab and pembrolizumab. We used the Xerna TME Panel, an investigational assay that identifies subtypes of the tumor microenvironment using an artificial neural network algorithm on the expression of 125 TME-related genes, to test the hypothesis that cancers with evidence of an enriched immune cell signature score (immune-high) are more likely to respond to bavituximab plus pembrolizumab compared to cancers with a low immune cell signature score (immune-low)[23]. RNA expression was analyzed from tumors of 19 patients with available pre-

treatment tumor tissue to classify TME subtypes. The overall response rates for this subset were very similar to the overall cohort (Supplementary Table 2). Immune-high patients had an objective response rate of 62.5%, which was significantly greater than the 9.1% ORR observed in the immune-low group (two-sided Fisher's exact, *p* = 0.04; Fig. 4a). The median PFS in the immune-high group was greater (median: 7.5 months; 95% CI, 1.1–13.8 months) compared to the immune-low group (median: 2.6 months; 95% CI, 0.4–4.6 months; Fig. 4b), but this difference did not reach statistical significance.

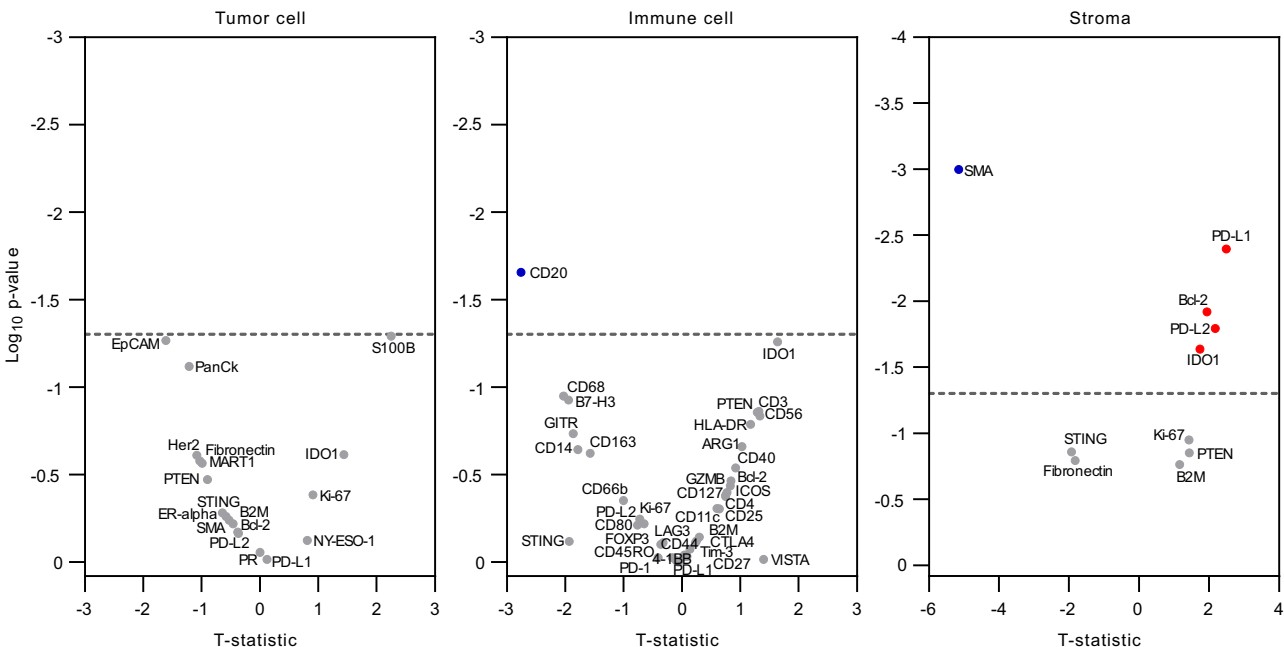

**Fig. 3 | Digital spatial profiling.** Volcano plots depict differentially expressed proteins across different cell compartments between patients with complete or partial response (*n* = 6) versus patients with stable or progressive disease (*n* = 14). Red dots represent proteins expressed at higher levels in responders while blude dots represent proteins expressed at higher levels in non-responders. Differential abundance of proteins according to objective response was determined by random permutation-based two-sided *t*-test. Dotted line indicates *p*-value threshold of 0.05. Source data are provided as a Source data file.

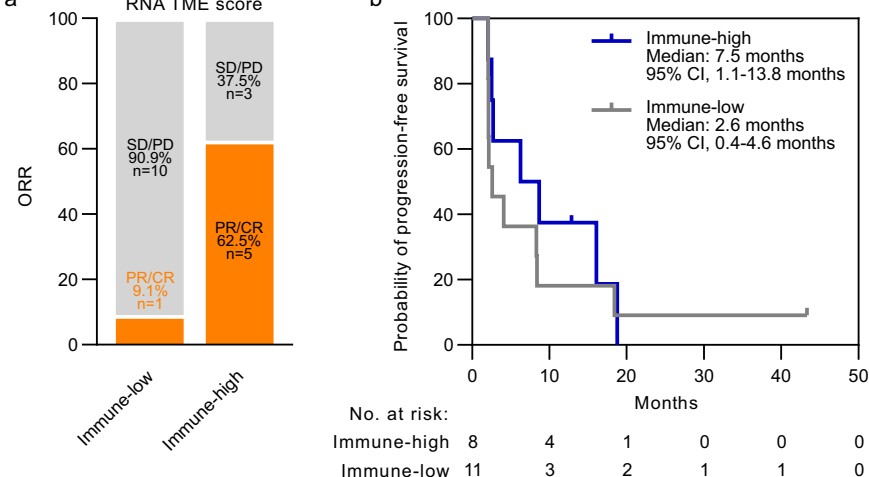

**Fig. 4 | Exploratory analysis of a RNA-based TME assay. a** Tumors were classified into as immune-low or immune-high based on the Xerna TME Panel. Stacked bar graph depicts ORR of each patient subset (two-sided Fisher's exact test *p* = 0.0408). ORR objective response rate, PR partial response, CR complete response, SD stable disease, PD progressive disease. **b** Kaplan-Meier curve shows the probability of progression-free survival over time among evaluable patients differentiated by Xerna TME biomarker status. Source data are provided as a Source data file.

## Discussion

To date, combination anti-PD-1/L1 regimens approved in HCC have either incorporated anti-VEGF or anti-CTLA-4 agents[2,3]. Whether other immunotherapy combinations are efficacious and safe in HCC remains under investigation. This study met its prespecified primary endpoint with a confirmed ORR of 32.1% and many responses being durable, suggesting that targeting phosphatidylserine may be an effective strategy of enhancing anti-PD-1/L1 activity in HCC. The observed ORR in this study is comparable to what has been observed with other doublet immunotherapy regimens in HCC (30% for atezolizumab plus bevacizumab in IMbrave150 and 20.1% for tremelimumab plus durvalumab in HIMALAYA)[2,3,24]. In addition, the ORR of bavituximab plus

pembrolizumab compares favorably with the ORR observed in Cohort 2 of the KEYNOTE-224 study where an ORR of 16% was observed in patients with advanced HCC treated in the first line setting with pembrolizumab monotherapy[14,15]. Nonetheless, our results require further validation in a large, randomized study because cross-trial comparisons are confounded by differences in study criteria, patient characteristics, statistical approaches, and other external factors. Notably, anti-VEGF agents are known to have monotherapy activity in HCC and other cancer types, bringing into question whether combining these drugs with anti-PD-1/L1 therapies leads to bona fide synergistic or additive effects[25,26]. In contrast, bavituximab has not shown meaningful monotherapy activity in multiple cancer types or

when combined with anti-VEGF therapies in HCC[7,10]. Thus, the ORR of bavituximab plus pembrolizumab may be the result of actual synergy, which warrants validation in future studies.

The safety profile of bavituximab plus pembrolizumab in this study also highlights the potential advantage of incorporating anti-phosphatidylserine with anti-PD-1/L1 therapies over existing frontline regimens for HCC. While atezolizumab plus bevacizumab is an effective frontline regimen in HCC, a proportion of patients with advanced HCC have other comorbidities including cardiovascular disease and diabetes which may increase their risk of complications[2,27]. In addition, an upper endoscopy is recommended to be performed prior to starting bevacizumab, which places additional clinical burden on patients and providers which may delay treatment. Consistent with its mechanism of action, bavituximab is not associated with an increased risk of adverse events related to endothelial injury, and the safety profile of bavituximab plus pembrolizumab is comparable to those reported in studies testing pembrolizumab monotherapy in HCC[14,16,17]. The rate of all and high-grade (grade 3 and 4) treatment-related adverse events observed with bavituximab plus pembrolizumab was also similar to rates reported for durvalumab monotherapy (all grades: 45.7% vs 52.1%; grade 3 and 4: 14.3% vs 12.9%), but substantially less than rates reported for tremelimumab plus durvalumab (all grade: 75.8%; grade 3 and 4: 25.8%)[3]. This suggests that combining bavituximab with pembrolizumab does not enhance toxicities associated with immune checkpoint inhibitors, and may be a better tolerated compared to existing doublet regimens.

This study also explored pre-treatment characteristics of the TME that may mediate anti-tumor immunity. For instance, the presence of infiltrating B cells and their inclusion in tertiary lymphoid structures in melanoma, sarcoma, and HCC treated with anti-PD-1/L1 therapies is associated with tumor response, suggesting that B cells may help orchestrate effective cancer rejection rejection[28-31]. We found that cancers responsive to bavituximab plus pembrolizumab were associated with low CD20 expression among immune cells. This may suggest that bavituximab may negate the requirement for intratumor B cells in the activation of effector lymphocytes and polarization of macrophages by ICIs[32]. Because phosphatidylserine and its receptors are broadly expressed among viable immune cell types in the TME, bavituximab is likely capable of inducing immunogenic responses involving multiple immune cell types and lineages, including the inhibition of regulatory B cells[33,34]. Consistent with a post-hoc analysis of the IMbrave150 trial where tumor and immune cell PD-L1 expression was not found to be associated with survival outcomes after atezolizumab plus bevacizumab in HCC, our proteomic analysis also did not find a relationship between tumor response to bavituximab plus pembrolizumab and PD-L1/2 expression in tumor or immune cells[35]. However, PD-L1, PD-L2, and IDO1 levels in stroma was associated with sensitivity to bavituximab plus pembrolizumab, suggesting that compartment specific expression of immunosuppressive molecules may be important mediators of ICI activity and potential biomarkers of response[36,37]. Notably, our analysis also indicates that increased SMA expression, reflecting intratumoral fibrosis and cancer-associated fibroblasts, may be associated with worse response to bavituximab plus pembrolizumab[38-40]. This finding is consistent with recent evidence that tumor-associated fibrosis in the lungs impairs immune surveillance and poor responsiveness to checkpoint blockade[41]. These findings may support the potential utility of combining anti-fibrotic therapies with ICIs in fibrotic HCC in future studies. Because our trial did not include a pembrolizumab monotherapy arm, it remains to be investigated whether TME characteristics associated with treatment response in this study are predictive of pembrolizumab monotherapy benefit as well.

This trial is limited by its small sample size, lack of blinding in radiographic assessments, and single-arm design. Thus, the generalizability of these results and the survival benefit of bavituximab plus pembrolizumab remains to be tested in a randomized setting. However, the encouraging results of this study lends support to future investigations testing anti-phosphatidylserine agents in combination with anti-PD-1/L1 therapies or existing frontline immunotherapy regimens. Notably, this trial enrolled patients at clinical sites targeting disparate demographics (a tertiary referral center and a safety net health system) and consequently had a high proportion of non-White patients analyzed. This highlights the feasibility of enrolling diverse demographics in HCC trials, which is critical given the long-standing underrepresentation of non-White race in trials[42].

## Methods

### Study design and endpoints

This study was an investigator-initiated phase 2, single-arm, open-label trial conducted at 2 clinical sites (UT Southwestern Harold C. Simmons Comprehensive Cancer Center and Parkland Health). The study was approved by the UT Southwestern institutional review board, and all patients provided written informed consent prior to enrollment. The study design and conduct complied with all relevant regulations regarding the use of human study participants and was conducted in accordance with the criteria set by the Declaration of Helsinki. This study was registered at Clinicaltrials.gov on April 26, 2018 (https://clinicaltrials.gov/ct2/show/NCT03519997). The trial protocol is provided in the Supplementary Information. There were no deviations that affected the trial design.

The primary endpoint was the confirmed ORR by Response Evaluation Criteria in Solid Tumors (RECIST) 1.1 of all evaluable patients receiving at least one dose of pembrolizumab and bavituximab. ORR was defined as number of patients with objective tumor responses (either CR or PR as best overall response) that were confirmed on imaging greater than or equal to 9 weeks after initial best response. A minimax 2-stage design was used to guide sample size determination. In the first stage, if 3 or more of 15 evaluable patients had objective responses, an additional 13 evaluable patients would be accrued. If 8 or more objective responses were observed in 28 evaluable patients, the study treatment would be considered worthwhile of further study. This sample size yields a type I error rate of 0.05 and power of 0.80 when the true response rate is 35% versus a historical control rate of 15% based on the response rate of single agent nivolumab in advanced HCC observed in the CheckMate-040 study. Secondary endpoints included OS, PFS, duration of response, and safety. Exploratory endpoints included correlative studies between treatment response and outcomes with clinical and laboratory studies. This study followed the Transparent Reporting of Evaluations With Nonrandomized Designs (TREND) reporting guideline, and was conducted in compliance with the trial protocol.

### Participants

Patients were enrolled from 25 June 2018 to 2 March 2022. Patients were eligible if they were 18 years or older and had a histologically confirmed diagnosis of HCC (excluding fibrolamellar, sarcomatoid, and combined subtypes), locally advanced or metastatic disease not amenable to surgical resection, transplantation, or locoregional therapies, Child-Pugh A disease, ECOG status of 0 or 1, a life expectancy greater than 6 months, adequate laboratory values (total bilirubin ≤2.0 mg/ml, international normalized ratio ≤1.7, hemoglobin ≥8.5 g/dl, aspartate transaminase and alanine transaminase ≤5 times upper limits of normal, platelet count ≥50,000/mm³, serum creatinine ≤1.5 mg/dL or creatinine clearance ≥ 50 mL/min, albumin ≥2.5 g/dl, absolute neutrophil ≥1500 cells/mm³), agreed to use an adequate method of contraception through the course of the study and 120 days after the last dose of study medication (male and female subjects of child bearing potential), and a negative pregnancy test if the subject was a woman of child bearing age. Patients with HBV were eligible if their infection was controlled (antiviral therapy for HBV must be given for at least

12 weeks and HBV viral load must be less than 100 IU/mL prior to first dose of study drug). Patients with active or resolved HCV infection as evidenced by detectable HCV RNA or antibody were also eligible. Prior surgical or locoregional therapy was allowed provided the following conditions were met: (1) at least 4 weeks since prior locoregional therapy including surgical resection, chemoembolization, radio-therapy, or ablation; (2) target lesion has increased in size by 25% or more or the target lesion was not treated with locoregional therapy.

Exclusion criteria included a prior liver transplant, prior systemic therapy for HCC, clinically significant, uncontrolled heart disease or cardiovascular within 12 months of screening date (presentation of acute coronary syndromes, coronary angioplasty or stenting, symptomatic pericarditis, New York Heart Association class III-IV heart failure, documented cardiomyopathy, left ventricular ejection fraction <40% as determined by MUGA scan or ECHO), known human immunodeficiency virus (HIV) positive, history of thromboembolic events (including both pulmonary embolism and deep venous thrombus but not including tumor thrombus) within the last 6 months, hypersensitivity to IV contrast and not able to take pre-medication, active fungal infections requiring systemic treatment within 7 days prior to screening, known history of or any evidence of interstitial lung disease or active noninfectious pneumonitis, poorly controlled hypertension which is defined as systolic blood pressure >150 mmHg or diastolic pressure >90 mmHg despite optimal medical management, pre-existing thyroid abnormality with thyroid function that cannot be maintained in the normal range with medication, autoimmune disease (with the exceptions of vitiligo, type I diabetes mellitus resolved childhood asthma or atopy, suspected autoimmune thyroid disorders if they are currently euthyroid or with residual hypothyroidism requiring only hormone replacement, and psoriasis requiring systemic therapy must be excluded from enrollment), any other concurrent severe and/or uncontrolled medical condition that would, in the investigator's judgment, cause unacceptable safety risks, contra-indicate patient participation in the study or compromise compliance with the protocol, known history of active bacillus tuberculosis, medical conditions requiring systemic treatment with either corticosteroids (>10 mg/day prednisone equivalent) or other immunosuppressive medications within 14 days of study administration (inhaled or topical steroids and adrenal replacement doses >10 mg/day prednisone equivalents are permitted in the absence of autoimmune disease), major surgery within 14 days prior to starting study drug or subject has not recovered from major side effects (tumor biopsy is not considered as major surgery), clinically apparent ascites on physical examination, known hypersensitivity to any of the excipients of bavituximab or pembrolizumab or monoclonal antibody, active gastrointestinal bleeding within previous 2 months, any condition requiring anti-platelet therapy (aspirin >300 mg/day, clopidogrel >75 mg/day), prisoners or subjects who are involuntarily incarcerated, symptomatic or clinically active brain metastases, pregnant (defined as state of a female after contraception and until the termination of gestation, confirmed by a positive hCG laboratory test) or nursing (lactating) women, prior immunotherapy including anti-PD-1, anti-PD-L1, or anti-PD-L2 agents; and dual active HBV infection (HBsAg (+) and/or detectable HBV DNA) and HCV infection (anti-HCV Ab(+) and detectable HCV RNA) at study entry.

## Procedures
Patients received 200 mg pembrolizumab every three weeks (up to 35 cycles) in combination with 3 mg/kg of bavituximab weekly until disease progression, discontinuation due to toxicity, withdrawal of consent, or study termination. The 3 mg/kg weekly dosing for bavituximab was used in this study as prior phase 1 studies on this drug were capped at the 3 mg/kg dose to provide a 5-fold safety margin for impairments in coagulation parameters and platelet function while maintaining drug concentrations

expected to be well in excess of the therapeutic range based on animal models[6]. The maximally tolerated dose for bavituximab was not reached in prior phase 1 studies testing bavituximab as monotherapy or in combination with chemotherapies. No dose reductions were allowed. Dose interruptions were permitted for immune-related toxicities, and treatment was resumed once the event resolved (grade 1 or less) and any corticosteroids used for toxicity management has been tapered. Pembrolizumab was permanently discontinued if immune-related adverse event did not resolve within 12 weeks of last corticosteroid dose or corticosteroids could not be reduced to ≤10 mg prednisone or equivalent per day within 12 weeks. Dose interruptions were also permitted in the case of medical/surgical events or logistical reasons not related to study therapy (e.g., elective surgery, unrelated medical events, patient vacation, and/or holidays) and treatment must have resumed within 6 weeks of the scheduled interruption. Prohibited concomitant medications or treatments during the screening and treatment phase of this trial included antineoplastic systemic chemotherapy or biological therapy, other immunotherapy agents, investigational agents other than pembrolizumab and bavituximab, radiation therapy, live vaccines, and systemic glucocorticoids for any purpose other than to modulate symptoms from an event of clinical interest of suspected immunologic etiology.

Response and progression were evaluated using RECIST 1.1 as determined by the investigator. Patients were evaluated by imaging for progression every 9 weeks for the first 54 weeks, then every 12 weeks thereafter following the initial study drug administration. Imaging consisted of computed tomography scans, or magnetic resonance imaging scans of the abdomen with triple-phase acquisition. CT Chest or additional imaging was at the discretion of the investigator based on clinical suspicion of additional progressive disease.

All patients receiving at least one dose of study drugs were evaluated in the safety analysis regardless of their evaluability for the primary ORR endpoint. Treatment-related adverse events were defined as any adverse event at least possibly attributed to study drugs per the protocol that occurred between the first dose of study drugs through 30 days after the last dose of study drugs. Adverse events occurring beyond 30 days at least possibly attributed to study drugs per the investigator was also reported. Adverse events were graded according to National Cancer Institute Common Terminology Criteria for AEs, v.5.0. All serious AE occurrences, attribution, and management were reviewed by the internal gastrointestinal oncology research committee composed of a multidisciplinary team of oncology providers and the UTSW Data and Safety Monitoring Committee.

## Clinical genomic sequencing
Clinical molecular profiling was performed using tissue-based sequencing assays (FoundationOne Cdx or Caris MI Tumorseek) on archival biopsy or resection specimens. When a tissue specimen was not accessible or insufficient, a circulating tumor DNA sequencing assay (Guardant 360) was used on a pre-treatment blood collection. For tissue-based sequencing, formalin-fixed, paraffin-embedded tissue biopsy or resection specimens were submitted to CLIA-certified and CAP-accredited laboratories (Foundation Medicine, Inc., Cambridge, MA or Caris Life Sciences, Phoenix, AZ) for hybrid capture-based next-generation sequencing (NGS)-based genomic profiling of 395 cancer-related genes, or NovaSeq-based sequencing of the exome. Somatic mutations were abstracted from clinical reports and genetic variants interpreted as variants of unknown significance, likely benign, or benign were filtered.

## Multiplex spatial profiling
The GeoMx Digital Spatial Profiler (NanoString) platform was used to assess spatial proteomics of baseline tissue specimens[43].

Immunofluorescent staining of formalin-fixed paraffin-embedded tumor tissues obtained from biopsy or resection specimens prior to treatment with bavituximab and pembrolizumab were performed using the following morphology markers for the segment selection: anti-SMA (Invitrogen, Catalog No. 50-9760-82, 1:40), anti-CD45 (Novus Biologicals, Catalog No. NBP1–44763AF594, 1:40), and anti-pan-cytokeratin (Pan-CK; GeoMx Solid Tumor TME Morphology Kit, NanoString, 1:40). Antibodies for 52 immuno-oncology-related proteins and three housekeeping/background proteins (S6, Histone H3, GAPDH, mouse IgG1, mouse IgG2a, rabbit IgG, 1:25) conjugated with photocleavable indexing oligonucleotides were hybridized to the tissues overnight. Three to five representative tumor-enriched regions of interest (ROI) were selected from tissue specimens from each patient using the Digital Spatial Profiler software. Within each ROI, areas enriched for cell types of interest were defined as "segments" based on the immunofluorescent staining, and abundance of the immuno-oncology-related proteins was measured in each segment as follows. Immune cell-enriched segments were defined as CD45-positive areas. Stroma-enriched segments were defined as SMA-positive areas. Tumor cell-enriched segments were defined as pan-CK-positive areas or areas lacking any of the three markers and enriched with tumor cells determined in H&E staining of serial tissue section. Raw abundance of each protein was measured by using the nCounter SPRINT Profiler (NanoString) and normalized by scaling with geometric mean of the housekeeping proteins using the GeoMx DSP system (NanoString). Differential abundance of proteins according to objective response was determined by random permutation-based $t$-test. Spatial profiling data are publicly available via the NCBI Gene Expression Omnibus database (accession number: GSE242154).

### Xerna TME panel analysis

Total RNA was extracted from pre-treatment formalin-fixed, paraffin-embedded (FFPE) tumor tissue samples using the All-Prep extraction kit (Qiagen) and quantified using RNA detection (Life Technologies). RNA-seq libraries were prepared using the Stranded total RNA Seq KAPA RNA HyperPrep Kit with RiboErase (Roche) from 100 ng of purified total RNA according to the manufacturer's protocol. The finished dsDNA libraries were quantified by Qubit fluorometer (Life Technologies), TapeStation 4200 (Agilent), and RT-qPCR using the Quant-iT High Sensitivity dsDNA assay kit (ThermoFisher Scientific) and Synergy 2 Multimode plate reader (Biotek) according to the manufacturer's protocols. Uniquely indexed libraries were pooled in equimolar ratios and sequenced on Illumina NextSeq500/550 sequencers with 2X76bp paired-end reads. Sequences were aligned to a human reference genome (Hg38) index using STAR v2.7.0. The reference genome index was built using STAR v2.7.0 with the Homo_sapiens GRCh38 genome and the Ensembl release 92 annotation. The resultant BAM files were saved for downstream processing. Per sample, sequencing depth was pulled from the aligned BAM file using a custom Python script. Read counts per gene were quantified from the aligned BAM files using Featurecounts v1.6.3 and the Ensembl release 92 genome annotation. Using the raw read counts, the coverage for each of 14 housekeeping (HK) genes were extracted and the median coverage calculated using a custom Python script. Per gene read counts were normalized using the rnanorm v1.0.0 Python package developed by Genialis.

The Xerna TME Panel identifies 4 subtypes for the tumor microenvironment using an artificial neural network algorithm of 125 genes to compute a score for two nodes that roughly correspond to "abnormal/pathological blood vessels" (or angiogenesis), and "immune" gene expression. Scores range from low (−1) to high (+1) for each of the angio (blood vessel) and immune scores. The four subtypes are characterized as: Angiogenesis (A), Immune Suppressed (IS), Immune Active (IA), or Immune Desert (ID). Data was analyzed with the Xerna TME Panel using previously described methods[44]. The clinical assay was run in a CLIA environment at Almac Diagnostics and is commercially available as part of the Oncomap ExTra diagnostic sold by Exact Sciences Inc. To assess associations between TME subtypes and tumor response or PFS after bavituximab and pembrolizumab, subtypes associated with a high immune score (IA and IS) were classified as immune-high, and subtypes associated with a low immune score (A and ID) were classified as immune-low.

### Statistical analysis

The primary endpoint of the study was investigator-assessed ORR by RECIST 1.1 criteria, and secondary endpoints reported were PFS and incidence of adverse events. ORR was defined as the percentage of patients with CRs and PRs that were subsequently confirmed on follow-up restaging scan. The disease control rate was defined as the percentage of patients with SD or confirmed CR/PRs as their best response per RECIST. PFS was estimated using the Kaplan-Meier method. 95% confidence intervals for the median PFS and 6-month PFS rate were calculated using the Greenwood method. Descriptive analyses were used to describe treatment-related adverse events. For exploratory analyses, Fisher's exact tests were used to assess associations between tumor response and molecular alterations or TME subtypes. Odds ratio confidence intervals were calculated using the Baptista-Pike method. Statistical analyses were performed using SPSS 28 (IBM); $p$-values were from 2-sided tests and results were deemed statistically significant at $p < 0.05$. Because correlative studies conducted were exploratory, analyses were not corrected for multiple-testing.

### Reporting summary

Further information on research design is available in the Nature Portfolio Reporting Summary linked to this article.

## Data availability

Individual de-identified clinical data may be requested through the corresponding author David Hsiehchen (David.hsieh@utsouthwestern.edu), which will require the approval of the institutional review board. Written requests from scientific investigators for research purposes that include data required and study aims will be responded to within 6 weeks. Data will be made accessible for a duration of 6 months. Processed data for spatial profiling data are publicly available via the NCBI Gene Expression Omnibus database with accession number GSE242154 and as Supplementary Data 1. Raw sequencing genomic data are protected by privacy laws and not publicly accessible because the study consent did not cover deposition of patient genetic data. Sequencing data cannot be made available upon request due to legal restrictions on disclosing patient information without explicit permission from individuals. The distribution of genes altered and specific molecular alterations among patients is provided in the Source data. The Xerna TME Panel analysis was performed by OncXerna using an assay licensed to Exact Sciences. Results from the assay was provided as a binary classification of which patients were immune-high or immune-low. Individual gene expression data are not available and cannot be shared due to proprietary information constraints. Remaining data are available in the Article and its Supplementary Information. Source data are provided with this paper.

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

## Acknowledgements

Merck and Co. Inc, supplied funding and pembrolizumab for the study. Oncxerna supplied bavituximab for this study. The funder had no role in study design, conduct, data collection, data analysis or the writing of this report. Y.H. is supported by the National Cancer Institute (CA233794, CA255621), European Commission (ERC-AdG-2020-101021417), and the Cancer Prevention & Research Institute of Texas (RR180016).

## Author contributions

M.S.B. and A.C.Y. conceived the study. D.H. and M.S.B. supervised the study. D.H., M.S.B., R.K., J.L., S.M.K., L.K., M.C.M., H.K., A.K.P., A.G.S., H.Z., and E.S. participated in patient enrollment and treatment. C.K., A.H., N.K., and Y.H. performed the NanoString analysis. H.Y., C.M., K.C., M.U., and L.B. performed the Xerna TME analysis. R.A.B. and C.A. contributed to the design of the study. D.H. drafted the manuscript. All authors participated in the revision of the manuscript and approved the paper.

## Competing interests

D.H. has served as a consultant for AztraZeneca. M.S.B. is an employee of Science37. H.Y., C.M., K.C., M.U., and L.B. are employees of OncXerna Therapeutics. A.S. has served as a consultant or on advisory boards for Genentech, AztraZeneca, Eisai, Exelixis, Bayer, Boston Scientific, Fuji-Film Medical Sciences, Exact Sciences, Roche, Glycotest, Universal Dx, Freenome, and GRAIL. Y.H. has served as a consultant or on advisory boards for Helio Genomics, Alentis Therapeutics, Espervita Therapeutics, and Roche Diagnostics, and holds shares in Alentis Therapeutics and Espervita Therapeutics. R.A.B. received research support from OncXerna. The remaining authors declare no competing interests.

## Additional information

**David Hsiehchen** [1,2,8] ✉, **Muhammad S. Beg**[1,2,8], **Radhika Kainthla**[1,2], **Jay Lohrey**[1,2], **Syed M. Kazmi**[1,2], **Leticia Khosama**[2], **Mary Claire Maxwell**[2], **Heather Kline**[2], **Courtney Katz** [2,3], **Asim Hassan**[2,3], **Naoto Kubota**[2,3], **Ellen Siglinsky**[2], **Anil K. Pillai**[2,4], **Hagop Youssoufian**[5], **Colleen Mockbee**[5], **Kerry Culm**[5], **Mark Uhlik**[5], **Laura Benjamin**[5], **Rolf A. Brekken** [2,6], **Chul Ahn**[2,7], **Amit G. Singal** [2,3], **Hao Zhu** [1,2], **Yujin Hoshida** [2,3] & **Adam C. Yopp**[2,6]

[1]Divison of Hematology and Oncology, Department of Internal Medicine, University of Texas Southwestern Medical Center, Dallas, TX, USA. [2]Harold C. Simmons Comprehensive Cancer Center, University of Texas Southwestern Medical Center, Dallas, TX, USA. [3]Divison of Digestive and Liver Disease, Department of Internal Medicine, University of Texas Southwestern Medical Center, Dallas, TX, USA. [4]Divison of Vascular and Interventional Radiology, Department of Radiology, University of Texas Southwestern Medical Center, Dallas, TX, USA. [5]OncXerna Therapeutics, Waltham, MA, USA. [6]Divison of Surgical Oncology, Department of Surgery, University of Texas Southwestern Medical Center, Dallas, TX, USA. [7]Peter O'Donnell Jr. School of Public Health, University of Texas Southwestern Medical Center, Dallas, TX, USA. [8]These authors contributed equally: David Hsiehchen, Muhammad S. Beg. ✉e-mail: David.hsieh@utsouthwestern.edu

