## [Peer Review File · Nature Communications]

The phosphatidylserine targeting antibody bavituximab plus pembrolizumab in unresectable hepatocellular carcinoma: a phase 2 trialREVIEWER COMMENTS

Reviewer #1 (Remarks to the Author): with expertise in biostatistics, clinical trial study design

This is an open-label single-arm Phase II trial conducted at two clinical sites, with the primary objective to determine the overall response rate of combination pembrolizumab and bavituximab in patients with advanced hepatocellular carcinoma. Secondary objectives included 6-month PFS, duration of response, overall survival and the safety and tolerability of the combination of pembrolizumab and bavituximab. Exploratory objectives included plasma/serum correlative studies and association of pre- and intra-treatment biopsies with treatment response.

Overall, this is an important study that provides preliminary findings on the activity of combination pembrolizumab and bavituximab in patients with advanced hepatocellular carcinoma.

Major comments:

1. The authors reported that there were 2 CRs and 7 PRs. In Figure 1a, there are however only 8 bars that extended below the 30% tumor shrinkage threshold line. Please explain why one of the patients who did not achieve at least a 30% decrease was considered a partial responder.
2. Secondary objectives listed in the protocol included 6-mth PFS. Although the median PFS was reported as 6.3 months, could the authors explain why they decided not to report the pre-specified secondary objective of 6-mth PFS in the manuscript?
3. Appreciate if the authors can include the median follow up time of this cohort of patients and state the method used to compute it in the statistical section.
4. Line 186, Supplementary Figure 2; Given that the endpoint is response, would have expected the estimates from the analysis to be odds ratios instead of hazard ratios, please check and confirm.
5. Supplementary Figure 2; It is not clear which category corresponds to the reference category for each patient demographics and baseline disease characteristics. Is the OR of response for No EHD against EHD (EHD as the reference) <1 while the OR of response for

male against female (female as the reference group) >1? It will be helpful to include the Odds ratios and 95% CI in the figure. Please include method used for this analysis in the statistical section.

6. Line 197-198, could the authors please explain why 1 patient discontinued therapy permanently per the treating investigator?

7. Line 204, should the percentage of CCNE1 be 9% or 11% (as stated in Figure 2a)

8. Could the authors please include the definition of ORR and DCR in the statistical section.

9. Could the authors please state the statistical method used to calculate the 95% CI for the median PFS (and/or 6-mth PFS) in the statistical section.

Reviewer #2 (Remarks to the Author): with expertise in HCC, (immuno)therapy

I have read with great interest the work from Hsiechen et al., and I would like to congratulate the authors for their huge effort. They put up a complex study, investigating the role of a drug with a novel mechanism of action in the increasingly intricate setting of 1st line treatment for advanced hepatocellular carcinoma.

I believe the main issue related to this study is the small sample size, which is rather limited for a Phase II study and does not allow robust comparisons with larger studies conducted in the same setting. I agree with the authors on the promising results in terms of ORR, however I would recommend more caution when discussing the results in view of the number of responders (N=9). Also, sadly the number of samples limits the breadth of the conclusions that can be drawn from the translational analyses, which are however scientifically sound.

Please find below some more specific comments.

1. I find the toxicity results very intriguing, especially for a combination treatment. However, it would be important to report on all the adverse events, alongside the treatment-related adverse events specifically. I believe this should be a general requirement for clinical trials, as it supports the reliability of the data collection. I would also add this to the abstract.

2. The rate of trAEs in this study is particularly low, and it is even lower than the percentages reported for ICI monotherapy in larger studies (ie durvalumab in the Himalaya trial, or

nivolumab in the CheckMate 459). Could the authors expand the discussion on this point?

3. The ORR results are certainly promising. However, when choosing the historical control rate for the power calculation, the authors opted for a fairly conservative 15%, borrowed by the Checkmate-040. A potential issue when choosing this historical control rate stems from the fact that the Checkmate-040 was conducted in a pretreated population, where we would expect a generally lower response than a treatment-naïve population such as the one enrolled in this study. Also, in more recent monotherapy cohorts, the response rates are higher (18.3% for pembrolizumab in the Keynote240, 17.0% for durvalumab in the Himalaya study).

4. It would be useful to know why the investigators did not design a preliminary Phase I study to assess the safety and efficacy of the combination of bavituximab plus pembrolizumab. Also, since bavituximab has not been tested as monotherapy in patients with HCC, I would recommend more caution when discussing its potential synergistic activity with pembrolizumab, as we simply do not know if it works as monotherapy for HCC.

5. In the discussion, I would recommend rephrasing the paragraph on the use of bevacizumab for patients with HCC. In particular, I would avoid stating that cardiovascular disease and diabetes preclude the use of bevacizumab, as this is the current standard of care treatment for advanced HCC and it is widely utilised in clinical practice also in patients with comorbidities. For the same reason, I would suggest avoiding the sentence in line 262, as I do not think there is current evidence supporting that “a sizeable proportion of patients with HCC are not eligible for atezolizumab plus bevacizumab”, and the references utilised (2,26) are not in line with this conclusion.

6. The translational analyses conducted on the samples collected during the trial are complex and well conducted. However, the small sample size impairs the solidity of the conclusions, especially regarding the genomic correlates of response, where I would avoid reporting on association “trends” (line 205), as this can also be entirely due to random effects. Same observation on “trend” applies to the paragraph on the TME Panel (PFS trend in lines 235-237). In regards of the spatial proteomic profiling, despite the intriguing results on the CD20 differential expression, I do not think that this finding can be attributed to an immune-modulating effect of bavituximab, as it is maybe too optimistically suggested in line 280 – sadly there is no data supporting this conclusion.

7. The authors conducted an exploratory analysis to investigate the correlation between a

number of clinical characteristics and ORR (Suppl Fig 2). I would suggest expanding the description of this analysis in the Methods, and I would also suggest considering a Cox-regression analysis for PFS, but I understand if the low sample size does not allow this.

8. A couple of minor observations: the final part of the introduction (lines 151-153) is not really part of the background and I believe it should belong to the results. Also, ORR of atezolizumab-bevacizumab is 30% as reported in the updated results of the IMbrave150 study (Cheng et al, J Hepatol 2022), and for this reason the percentage of 27.3% reported in line 245 should be amended and the reference should be updated. Finally, CTLA-4 as monotherapy is not known to have a significant anti-cancer effect in HCC, and I would suggest rephrasing line 250.

Reviewer #3 (Remarks to the Author): with expertise in HCC, (immuno)therapy

This is a single arm study of bavituximab (targeting phosphatidylserine) in combination with pembrolizumab as first line systemic therapy for patients with advanced HCC and preserved liver function and performance status.

The scientific rationale for the study is sound.

The primary outcome measure was objective radiological response rate.

Of 28 evaluable patients, 9 objective responses were reported (ORR 32.1%), exceeding the prespecified threshold of 8 of 28 responses.

This response rate is encouraging when compared with historical data of response rates for ICI monotherapy and is similar to response rates reported for other immunotherapy combinations in HCC.

The toxicity profile also appears encouraging.

A number of correlative studies are reported, suggesting potential trends for higher response rates in some subgroups, including those with mutant p53; wild type CTNNB1; low B-cell number; high PD-L1, high PD-L2, high IDO1, high BCL2; and an immune high RNA signature in the TME.

Despite these encouraging data, there are a number of limitations and concerns:

1. As acknowledged by the authors, the study is limited by its small size and being a single arm study.
2. Of 35 patients enrolled, only 28 were evaluable. The reasons for non-evaluability of the

other 7 patients is only briefly described and requires more detail: two patients withdrew consent - why? could this have been due to clinical deterioration, suggesting non-response? two patients were withdrawn due to use of a prohibited concomitant medication - what medication and did these patients have any response assessment before withdrawal? two patients were withdrawn due to 'worsening concurrent illness' - could this have been related to HCC progression, again suggesting possible non-responders?

3. Was response assessment performed in an unblinded manner? If so, the potential bias of this should be stated.

4. Similarly, the unblinded nature of adverse event reporting should be stated particularly as a potential limitation in comparing frequency of adverse events across other trials.

5. Indeed, much cross-trial comparison is made and the limitations/biases associated with this should be stated

6. The text reports responses in 9 of 28 evaluable patients. However, the waterfall plot in fig 1a suggests responses in only 8 of 28. Can this discrepancy be explained.

7. The conclusions of the correlative studies are overstated. Much is made of 'trends' towards higher response rate or longer PFS in some subgroups. However, the study is underpowered for these multiple analyses and none are statistically significant. Further, the authors claim these may be predictive of benefit from the combination of bavituximab+pembrolizumab. However, the single arm nature of the study is such that they may be predictive of response to pembrolizumab alone, rather than the combination. This should also be stated.

8. No analysis of the extracellular expression of phosphatidylserine on tumour cells or within the TME. This is disappointing given this is the target of bavituxumab. Can the authors comment on this?

Reviewer #1 (Remarks to the Author): with expertise in biostatistics, clinical trial study design

This is an open-label single-arm Phase II trial conducted at two clinical sites, with the primary objective to determine the overall response rate of combination pembrolizumab and bavituximab in patients with advanced hepatocellular carcinoma. Secondary objectives included 6-month PFS, duration of response, overall survival and the safety and tolerability of the combination of pembrolizumab and bavituximab. Exploratory objectives included plasma/serum correlative studies and association of pre- and intra-treatment biopsies with treatment response.

Overall, this is an important study that provides preliminary findings on the activity of combination pembrolizumab and bavituximab in patients with advanced hepatocellular carcinoma.

Author response: We appreciate the reviewer's interest and constructive comments.

Major comments:

1. The authors reported that there were 2 CRs and 7 PRs. In Figure 1a, there are however only 8 bars that extended below the 30% tumor shrinkage threshold line. Please explain why one of the patients who did not achieve at least a 30% decrease was considered a partial responder.

Author response: We apologize for this error in the figure. The % tumor shrinkage in the plot was supposed to represent the maximal tumor shrinkage at any time, but for one patient it was inadvertently based on changes in tumor dimensions at their first imaging assessment. We have now corrected the figure.

2. Secondary objectives listed in the protocol included 6-mth PFS. Although the median PFS was reported as 6.3 months, could the authors explain why they decided not to report the pre-specified secondary objective of 6-mth PFS in the manuscript?

Author response: We apologize for this oversight and have now included the 6-m PFS in our results: "The 6-month progression-free survival (PFS) rate was 57.1% (95% CI 38.8-75.4)"

3. Appreciate if the authors can include the median follow up time of this cohort of patients and state the method used to compute it in the statistical section.

Author response: We now state in the Results: "The median follow-up time was 28.5 months using the reverse Kaplan-Meier method."

4. Line 186, Supplementary Figure 2; Given that the endpoint is response, would have expected the estimates from the analysis to be odds ratios instead of hazard ratios, please check and confirm.

Author response: We appreciate the reviewer's attention and have corrected this error.

5. Supplementary Figure 2; It is not clear which category corresponds to the reference category for each patient demographics and baseline disease characteristics. Is the OR of response for No EHD against EHD (EHD as the reference) <1 while the OR of response for male against female (female as the reference group) >1? It will be helpful to include the Odds ratios and 95% CI in the figure. Please include method used for this analysis in the statistical section.

Author response: We agree that the data is presented ambiguously and have now updated it so the reference group is clearly indicated. We have also added the 95% CI as recommended which was calculated using the Baptista-Pike method.

6. Line 197-198, could the authors please explain why 1 patient discontinued therapy permanently per the treating investigator?

Author response: We now clarify that treatment was discontinued due to grade 4 diarrhea which was a protocol requirement (due to a suspected high grade immune related adverse event). We have now revised the text to state: "1 patient discontinued therapy permanently for grade 4 diarrhea per the trial protocol for suspected high grade immune-related toxicity."

7. Line 204, should the percentage of CCNE1 be 9% or 11% (as stated in Figure 2a)

Author response: We appreciate the reviewer's attention and have corrected this to 11%.

8. Could the authors please include the definition of ORR and DCR in the statistical section.

Author response: We have now added the following language in the methods: "ORR was defined as the percentage of patients with CRs and PRs that were subsequently confirmed on follow-up restaging scan. The disease control rate was defined as the percentage of patients with SD or confirmed CR/PRs as their best response per RECIST."

9. Could the authors please state the statistical method used to calculate the 95% CI for the median PFS (and/or 6-mth PFS) in the statistical section.

Author response: We have now added the following language in the methods: "95% confidence intervals for the median PFS and 6-month PFS rate were calculated using the Greenwood method."

Reviewer #2 (Remarks to the Author): with expertise in HCC, (immuno)therapy

I have read with great interest the work from Hsiechen et al., and I would like to congratulate the authors for their huge effort. They put up a complex study, investigating the role of a drug with a novel mechanism of action in the increasingly intricate setting of 1st line treatment for advanced hepatocellular carcinoma.

I believe the main issue related to this study is the small sample size, which is rather limited for a Phase II study and does not allow robust comparisons with larger studies conducted in the same setting. I agree with the authors on the promising results in terms of ORR, however I would recommend more caution when discussing the results in view of the number of responders (N=9). Also, sadly the number of samples limits the breadth of the conclusions that can be drawn from the translational analyses, which are however scientifically sound.

Author response: We appreciate the reviewer's interest and agree with their concerns which we have made revisions to address.

Please find below some more specific comments.

1. I find the toxicity results very intriguing, especially for a combination treatment. However, it would be

important to report on all the adverse events, alongside the treatment-related adverse events specifically. I believe this should be a general requirement for clinical trials, as it supports the reliability of the data collection. I would also add this to the abstract.

Author response: As recommended, we have now included the following data in the results: “35 patients included in the safety analysis, adverse events of any cause were observed in 33 patients (94.3%) and serious adverse events of any cause were observed in 7 patients (20%). Adverse events of any cause that were grade 3 or greater occurred in 11 patients (31.4%). Treatment related adverse events occurred in 16 patients (45.7%) and serious adverse events attributed to study treatment occurred in 4 patients (11.4%).” We have also updated the abstract to include this data.

2. The rate of trAEs in this study is particularly low, and it is even lower than the percentages reported for ICI monotherapy in larger studies (ie durvalumab in the Himalaya trial, or nivolumab in the CheckMate 459). Could the authors expand the discussion on this point?

Author response: We are uncertain of the reviewer’s point because we believe that AE rates observed with pembro bavi were not meaningfully different from past monotherapy trials, but do seem less than combination trials. We summarize TRAE for all and high grades for our study and past monotherapy trials below:

**Pembro plus bavi: 45.7% all grade, 14.3% >G3
HIMALAYA, durvalumab: 52.1% all grade, 12.9% >G3
KEYNOTE-394, pembrolizumab: 66.9%, 13.3% >G3
KEYNOTE-224, pembrolizumab: 44%, 16% >G3**

The Lancet Onc paper describing the primary analysis of the CheckMate 459 trial does not seem to provide the unique number of patients who had TRAEs of any grade. They only provide it broken down by grades, but given that patients may have both G1, 2, 3, 4, we cannot sum the provided numbers to infer the actual TRAE rate.

3. The ORR results are certainly promising. However, when choosing the historical control rate for the power calculation, the authors opted for a fairly conservative 15%, borrowed by the Checkmate-040. A potential issue when choosing this historical control rate stems from the fact that the Checkmate-040 was conducted in a pretreated population, where we would expect a generally lower response than a treatment-naïve population such as the one enrolled in this study. Also, in more recent monotherapy cohorts, the response rates are higher (18.3% for pembrolizumab in the Keynote240, 17.0% for durvalumab in the Himalaya study).

Author response: This is a good point to which we are happy to provide more context. This study was designed between 2016-2017, and the only reference at that time was the results from the Checkmate-040 published in 2017. KEYNOTE-240 was published in 2018, by which time our trial was activated and enrolling patients. We agree that there seems to be a range of ORR observed between drugs and studies, and in hindsight a 15% null rate does appear conservative. However, we also note that the ORR in contemporary studies including CheckMate 459 and KEYNOTE-394 had ORRs of 15% and 12.7%, respectively.

4. It would be useful to know why the investigators did not design a preliminary Phase I study to assess the safety and efficacy of the combination of bavituximab plus pembrolizumab. Also, since bavituximab

has not been tested as monotherapy in patients with HCC, I would recommend more caution when discussing its potential synergistic activity with pembrolizumab, as we simply do not know if it works as monotherapy for HCC.

Author response: Bavituximab has been tested in several phase 1 studies including combination studies with multiple chemotherapy agents. In all phase 1 studies including the combination studies, the MTD for bavituximab was not reached, with the highest dose of 3 mg/kg weekly. The phase 1 studies were capped at the 3 mg/kg dose to provide a 5-fold safety margin for impairments in coagulation parameters and platelet function while maintaining drug concentrations expected to be well in excess of the therapeutic range based on animal models. As stated in our Introduction, no prior monotherapy trials of bavituximab have demonstrated convincing single-agent activity in any cancer, and later-stage studies have not shown that bavituximab enhances chemotherapy outcomes. In addition, in our prior IST which tested sorafenib plus bavituximab, the observed orr was nearly identical to the orr reported in SHARP. These points argue against bavituximab having significant single agent activity in cancer including HCC.

Our introduction briefly covers the initial phase 1 and the subsequent phase 2 study in HCC. We also summarize the above-mentioned information in the Methods Procedure section. We believe that a detailed history of the clinical development of bavituximab is beyond the scope of this paper but would likely be of interest to other readers, so we have now cited in our introduction and methods a review that covers this: Immunotherapy. 2011 Aug;3(8):933-44.

5. In the discussion, I would recommend rephrasing the paragraph on the use of bevacizumab for patients with HCC. In particular, I would avoid stating that cardiovascular disease and diabetes preclude the use of bevacizumab, as this is the current standard of care treatment for advanced HCC and it is widely utilised in clinical practice also in patients with comorbidities. For the same reason, I would suggest avoiding the sentence in line 262, as I do not think there is current evidence supporting that “a sizeable proportion of patients with HCC are not eligible for atezolizumab plus bevacizumab”, and the references utilised (2,26) are not in line with this conclusion.

Author response: We shared this point because the exclusion criteria for IMbrave150 was fairly restrictive and included recent cardiac disease/events within 3 months, vascular events within 6 months, being on full dose aspirin, and BPs greater than 150 SBP or 100 DBP. In addition, proteinuria was reported in up to 20% of patients in the trial, and at our institution, this is higher among patients with diabetes. Nonetheless, we agree with the reviewer’s point that atezo bev is likely used in clinical practice beyond the parameters of the initial trial population. We have now modified this sentence: “While atezolizumab plus bevacizumab is an effective frontline regimen in HCC, a proportion of patients with advanced HCC have other comorbidities including cardiovascular disease and diabetes which may increase their risk of complications.”

6. The translational analyses conducted on the samples collected during the trial are complex and well conducted. However, the small sample size impairs the solidity of the conclusions, especially regarding the genomic correlates of response, where I would avoid reporting on association “trends” (line 205), as this can also be entirely due to random effects. Same observation on “trend” applies to the paragraph on the TME Panel (PFS trend in lines 235-237). In regards of the spatial proteomic profiling, despite the intriguing results on the CD20 differential expression, I do not think that this finding can be attributed to an immune-modulating effect of bavituximab, as it is maybe too optimistically suggested in line 280 –

sadly there is no data supporting this conclusion.

Author response: The reviewer's comment is well taken. We have tempered our language and state that there are no statistically significant alterations associated with ORR. Similarly, we now state that the PFS between patients stratified by RNA signatures are not statistically significant.

7. The authors conducted an exploratory analysis to investigate the correlation between a number of clinical characteristics and ORR (Suppl Fig 2). I would suggest expanding the description of this analysis in the Methods, and I would also suggest considering a Cox-regression analysis for PFS, but I understand if the low sample size does not allow this.

Author response: We have now expanded on the specific statistical analyses in our Methods section. Due to several subgroups such as sex and MVI having very imbalanced sample sizes, and multicollinearity, regression models are likely to be unstable or unreliable.

8. A couple of minor observations: the final part of the introduction (lines 151-153) is not really part of the background and I believe it should belong to the results. Also, ORR of atezolizumab-bevacizumab is 30% as reported in the updated results of the IMbrave150 study (Cheng et al, J Hepatol 2022), and for this reason the percentage of 27.3% reported in line 245 should be amended and the reference should be updated. Finally, CTLA-4 as monotherapy is not known to have a significant anti-cancer effect in HCC, and I would suggest rephrasing line 250.

Author response: We appreciate these recommendations and have made several corrections including the removal of the sentence in the introduction, updating the ORR rate from IMbrave150 in the discussion, and removing the reference to anti-CTLA-4 monotherapy.

Reviewer #3 (Remarks to the Author): with expertise in HCC, (immuno)therapy

This is a single arm study of bavituximab (targeting phosphatidylserine) in combination with pembrolizumab as first line systemic therapy for patients with advanced HCC and preserved liver function and performance status.

The scientific rationale for the study is sound.

The primary outcome measure was objective radiological response rate.

Of 28 evaluable patients, 9 objective responses were reported (ORR 32.1%), exceeding the prespecified threshold of 8 of 28 responses.

This response rate is encouraging when compared with historical data of response rates for ICI monotherapy and is similar to response rates reported for other immunotherapy combinations in HCC. The toxicity profile also appears encouraging.

A number of correlative studies are reported, suggesting potential trends for higher response rates in some subgroups, including those with mutant p53; wild type CTNNB1; low B-cell number; high PD-L1, high PD-L2, high IDO1, high BCL2; and an immune high RNA signature in the TME.

Author response: We thank the reviewer for their interest.

Despite these encouraging data, there are a number of limitations and concerns:

1. As acknowledged by the authors, the study is limited by its small size and being a single arm study.

Author response: We agree with this as the main limitation of the study and have highlighted this in our discussion.

2. Of 35 patients enrolled, only 28 were evaluable. The reasons for non-evaluability of the other 7 patients is only briefly described and requires more detail: two patients withdrew consent - why? could this have been due to clinical deterioration, suggesting non-response? two patients were withdrawn due to use of a prohibited concomitant medication - what medication and did these patients have any response assessment before withdrawal? two patients were withdrawn due to 'worsening concurrent illness' - could this have been related to HCC progression, again suggesting possible non-responders?

Author response: We are happy to provide additional details on these patients, though precise reasons were not specifically captured for all patients as a part of this study. None of the unevaluable 7 patients had radiographic assessment of their disease prior to their removal off study. Of the 2 patients that withdrew consent, both were otherwise clinically well at their last visit. Records indicate that one of the patients subsequently received care locally due to transportation burdens, while the other patient sought care at a different hospital system. Of the 2 patients that were withdrawn due to prohibited treatment, one received palliative radiation to a pelvic metastasis which was a target lesion prior to C2. The second patient was removed from the trial due to intermittent hypoglycemia thought to be related to non-islet cell hypoglycemia. The patient had a prior recent history of hypoglycemia episodes but had normal BS on his screening visit and first date of infusion. 4 days after his first day of treatment, the patient was admitted for severe hypoglycemia with work up suspicious for a paraneoplastic process rather than an irAE. The patient was taken off study due to his hypoglycemia persisting. We have corrected the attribution of this patient to the patient being withdrawn due to concomitant illness. Of the remaining 2 patients, 1 patient developed high grade colitis during C1 for which she was removed off study per protocol, and the other patient developed hepatic encephalopathy and hepatorenal syndrome during cycle 1 which was determined by our GI committee and DSMC to be unrelated to treatment.

We have revised our results section to include the aforementioned details. However, it is speculative whether these patients would be considered non-responders, so we have intentionally not attempted to discuss this.

3. Was response assessment performed in an unblinded manner? If so, the potential bias of this should be stated.

Author response: We now clarify that ORR was investigator determined, and now include this as a limitation of this study.

4. Similarly, the unblinded nature of adverse event reporting should be stated particularly as a potential limitation in comparing frequency of adverse events across other trials.

Author response: We are unsure of the reviewer's concern, as an assessment particularly of any open-label and interventional study would face the same issue. We do not believe that comparisons of AE rates to IMbrave150 and HIMALAYA are unreasonable given that both studies were open-label studies.

We wish to note that per our institutions procedures and the trial protocol, all serious AEs are reviewed by the internal gastrointestinal oncology research committee and the DSMC so that

attribution and management of AEs are based on a consensus of a multidisciplinary team of oncology providers. We now include this detail in our methods.

5. Indeed, much cross-trial comparison is made and the limitations/biases associated with this should be stated

Author response: We agree with the reviewer's point. We now state in our Discussion: "Nonetheless, our results require further validation in a large, randomized study because cross-trial comparisons are confounded by differences in study criteria, patient characteristics, statistical approaches, and other external factors."

6. The text reports responses in 9 of 28 evaluable patients. However, the waterfall plot in fig 1a suggests responses in only 8 of 28. Can this discrepancy be explained.

Author response: We apologize for this error in the figure. The % tumor shrinkage in the plot was supposed to represent the maximal tumor shrinkage at any time, but for one patient it was inadvertently based on changes in tumor dimensions at their first imaging assessment. We have now corrected the figure.

7. The conclusions of the correlative studies are overstated. Much is made of 'trends' towards higher response rate or longer PFS in some subgroups. However, the study is underpowered for these multiple analyses and none are statistically significant. Further, the authors claim these may be predictive of benefit from the combination of bavituximab+pembrolizumab. However, the single arm nature of the study is such that they may be predictive of response to pembrolizumab alone, rather than the combination. This should also be stated.

Author response: We agree with this point and have removed the reporting of trend associations and simply state that no statistical associations were identified. We also now clarify that our biomarkers do not distinguish benefit between pembro alone or pembro plus bavi: "Because our trial did not include a pembrolizumab monotherapy arm, it remains to be investigated whether TME characteristics associated with treatment response in this study are predictive of pembrolizumab monotherapy benefit as well."

8. No analysis of the extracellular expression of phosphatidylserine on tumour cells or within the TME. This is disappointing given this is the target of bavituxumab. Can the authors comment on this?

Author response: This is a good point. Unfortunately, there is not an easy way to assess phosphatidylserine externalization in tissue through conventional methods such as IHC because this does not distinguish phosphatidylserine on the outer vs inner leaflet of the plasma membrane. In vivo localization is the most effective way to see externalized phosphatidylserine, but is not possible in our patients/archival specimens.

REVIEWERS' COMMENTS

Reviewer #1 (Remarks to the Author):

The authors have satisfactorily addressed most of my comments, however, I could not find Supplementary Figure 2 in the revised manuscript. As such, am unable to review the revisions that the authors have made in relation to comments 4 and 5.

Reviewer #2 (Remarks to the Author):

I would like to thank the authors for addressing the reviewers' comments.

I believe the quality of the manuscript has now substantially improved, and I only have a further observation to make.

1. I appreciate the addition of the data regarding the overall adverse events. I would suggest adding to the methods the definition of "serious adverse events". Does this refer to AEs >G2? Also, I would recommend being consistent on the use of "serious AEs" vs "severe AEs", as in the manuscript the two terms are used interchangeably.

Reviewer #3 (Remarks to the Author):

The authors have addressed the reviewer comments comprehensively, thank you

Reviewer #1 (Remarks to the Author):

The authors have satisfactorily addressed most of my comments, however, I could not find Supplementary Figure 2 in the revised manuscript. As such, am unable to review the revisions that the authors have made in relation to comments 4 and 5.

Author response: We have ensured that Supp Fig 2 is now in the Supp file.

Reviewer #2 (Remarks to the Author):

I would like to thank the authors for addressing the reviewers' comments. I believe the quality of the manuscript has now substantially improved, and I only have a further observation to make.

1. I appreciate the addition of the data regarding the overall adverse events. I would suggest adding to the methods the definition of "serious adverse events". Does this refer to AEs >G2? Also, I would recommend being consistent on the use of "serious AEs" vs "severe AEs", as in the manuscript the two terms are used interchangeably.

Author response: We appreciate the reviewer's comment. We have now included our trial protocol which provides a detailed description of serious AEs. We have also edited the paper to change the reference to severe AEs to high-grade AEs.

Reviewer #3 (Remarks to the Author):

The authors have addressed the reviewer comments comprehensively, thank you

Author response: We appreciate the reviewer's interest.